# Nutrients Supplementation through Organic Manures Influence the Growth of Weeds and Maize Productivity

**DOI:** 10.3390/molecules25214924

**Published:** 2020-10-24

**Authors:** Dibakar Ghosh, Koushik Brahmachari, Milan Skalicky, Akbar Hossain, Sukamal Sarkar, Nirmal Kumar Dinda, Anupam Das, Biswajit Pramanick, Debojyoti Moulick, Marian Brestic, Muhammad Ali Raza, Celaleddin Barutcular, Shah Fahad, Hirofumi Saneoka, Ayman EL Sabagh

**Affiliations:** 1ICAR-Directorate of Weed Research, Jabalpur, Madhya Pradesh 482004, India; dghoshagro@gmail.com; 2Department of Agronomy, Bidhan Chandra Krishi Viswavidyalaya, Mohanpur, Nadia, West Bengal 741252, India; brahmacharis@gmail.com (K.B.); sukamalsarkarc@yahoo.com (S.S.); 3Department of Botany and Plant Physiology, Faculty of Agrobiology, Food and Natural Resources, Czech University of Life Sciences Prague, Kamycka 129, 16500 Prague, Czech Republic; marian.brestic@uniag.sk; 4Bangladesh Wheat and Maize Research Institute, Dinajpur 5200, Bangladesh; 5Office of the Assistant Director of Agriculture, Suri II Block, Department of Agriculture, Government of West Bengal, Birbhum, West Bengal 731129, India; nirmaldinda@gmail.com; 6Department of Soil Science and Agricultural Chemistry, Bihar Agricultural University, Bhagalpur, Bihar 813210, India; anusoil22@gmail.com; 7Department of Agronomy, Dr. Rajendra Prasad Central Agricultural University, Pusa, Samastipur, Bihar 848125, India; bipra.its4u@gmail.com; 8Plant Stress Biology and Metabolomics Laboratory, Central Instrumentation Laboratory, Assam University, Silchar, Assam 788011, India; drubha31@gmail.com; 9Department of Plant Physiology, Slovak University of Agriculture, Nitra, Tr. A. Hlinku 2, 94901 Nitra, Slovakia; 10College of Agronomy, Sichuan Agricultural University, Chengdu 611130, China; razaali0784@yahoo.com; 11Department of Field Crops, Faculty of Agriculture, University of Çukurova, Sarıçam/Adana 01330, Turkey; cbarutcular@gmail.com; 12Department of Agronomy, the University of Haripur, Khyber Pakhtunkhwa 22620, Pakistan; shahfahad@uoswabi.edu.pk; 13Graduate School of Integrated Sciences for Life, Hiroshima University, 1-4-4 Kagamiyama, Higashi Hiroshima 739-8528, Japan; saneoka@hiroshima-u.ac.jp; 14Department of Field Crops, Faculty of Agriculture, Siirt University, Siirt 56100, Turkey; aymanelsabagh@gmail.com; 15Department of Agronomy, Faculty of Agriculture, Kafrelsheikh University, Kafr El-Shaikh 33516, Egypt

**Keywords:** nutrient management, weed management, *Brassicaceous* seed meal, neem cake, herbicide, nutrient uptake

## Abstract

Declining rate of productivity and environmental sustainability is forcing growers to use organic manures as a source of nutrient supplement in maize farming. However, weed is a major constraint to maize production. A field study was carried out over two seasons to evaluate various integrated nutrient and weed management practices in hybrid maize. The treatment combinations comprised of supplementation of inorganic fertilizer (25% nitrogen) through bulky (Farmyard manure and vermicompost) and concentrated (*Brassicaceous* seed meal (BSM) and neem cake (NC)) organic manures and different mode of weed management practices like chemical (atrazine 1000 g ha^−1^) and integrated approach (atrazine 1000 g ha^−1^ followed by mechanical weeding). Repeated supplementation of nitrogen through concentrated organic manures reduced the density and biomass accumulation of most dominant weed species, *Anagalis arvensis* by releasing allelochemicals into the soil. But organic manures had no significant impact on restricting the growth of bold seeded weeds like *Vicia hirsuta* and weed propagated through tubers i.e., *Cyperus rotundus* in maize. By restricting the weed growth and nutrient removal by most dominating weeds, application of BSM enhanced the growth and yield of maize crop. Repeated addition of organic manures (BSM) enhanced the maize grain yield by 19% over sole chemical fertilizer in the second year of study. Application of atrazine as pre-emergence (PRE) herbicide significantly reduced the density of *A*. *arvensis*, whereas integration of mechanical weeding following herbicide controlled those weeds which were not usually controlled with the application of atrazine. As a result, atrazine at PRE followed by mechanical weeding produced the highest maize grain yield 6.81 and 7.10 t/ha in the first year and second year of study, respectively.

## 1. Introduction

Maize (*Zea mays* L.) is the third most important cereal crop across the globe. The importance of maize lies in its wide range of industrial uses, besides serving as human food and animal feed. It can be grown throughout the year due to its photo-insensitive nature and the highest genetic yield potential [1]. Maize, together with rice and wheat, make up at least 30% of the food demand to people of 94 developing countries [2]. In India, maize is grown in 9.2 million ha, with an annual production of about 24 M tons [3].

Maize exhausts a considerably higher amount of nutrients from soil as compared to other cereals [4]. Plant nutrition management is the pivotal factor that can regulate the crop yield. But, the inherent fertility status of the soil greatly varies spatially and temporally from the field to regional scale [5]. Therefore, development of a location-specific and balanced plant nutrient management system is highly required to maintain the soil health without compromising crop productivity. Under an intensive production system, incessant use of high-analysis chemical fertilizers causes rapid depletion in the soil organic matter, and increases soil acidity and environment pollution, resulting in a decline in total factor productivity as well as soil health [6,7]. Integrated nutrient management combines both traditional and modern methods of nutrient management for the development of the ecologically sound and economically optimal farming system. Integrated nutrient management enhances crop yields by about 8–15% as compared to the conventional practices [8]. Adoption of appropriate integrated nutrient management strategies not only increases the economic returns to farmers but also improves the soil health and environmental sustainability.

On the other hand, after plant nutrition, weeds are the second most limiting factor for efficient maize production. In uncontrolled conditions, weeds can cause up to 80% yield reduction in maize production systems [9]. The dynamic nature of weeds requires continuous development of effective weed management strategies by apposite combinations of mechanical, chemical, and agronomic strategies [10]. Amongst the various weed management strategies, chemical weed management gained widespread acceptability amongst farmers due to the ease of application, low cost involved, and effectiveness [11]. However, an acute rise in the price of herbicides, as well as emerging concerns about environmental pollution due to indiscriminate use of herbicides, have imposed to shift towards integrated weed management practices.

Weed flora possesses a more competitive relationship with applied plant nutrients than that of crops because nutrient absorption in weeds is often faster and higher than in crop plants [11]. Thus, plant nutrient management may play a significant role in regulating weed interference in crops. The “4R nutrient stewardship concept” (Right source at the Right rate, Right time, and Right place; https://nutrientstewardship.org/4rs/) of nutrient management may affect weed emergence, persistence, distribution, dynamics, as well as growth and yield attributes of crops [12]. Moreover, in integrated nutrients management (INM) practice, the relationship between crop–weed–plant nutrients became more complex because the organic source of plant nutrients may exhibit a synergistic [11] or antagonistic [13] relationship with weeds. The addition of farmyard manure and vermicompost as nutrient sources may enhance or diminish the weed growth [14,15,16,17,18], whereas *Brassicaceous* seed meal [19,20,21,22] and neem cake [23,24] having an allelopathic impact on weed seed germination.

For exploiting the full potential of hybrid maize, it is necessary to evaluate the complex interrelationship between crop and weed flora as influenced by integrated weed and nutrient management practices. A quantitative understanding of the crop response to imposed nutrient and weed management is crucial to formulate integrated weed and nutrient management practices. Reports on crop performance, macronutrient removal by different weed flora, as well as the interrelationship with associated maize crop in the Lower Indo-Gangetic alluvial zone are limited. Thus, the present study aimed to formulate a sustainable maize production system through integrated weed and nutrient management practices in the alluvial plains of West Bengal. The explicit objectives were: (i) to study the effect of different nutrient sources and weed management practices on growth and productivity of maize, and (ii) to identify the different weed flora and assess their growth, nutrient removal, and interrelationship with crop under varied treatment combinations.

## 2. Results

### 2.1. Weed Density

The weed flora of the experimental site were *Anagallis arvensis* L., *Cyperus rotundus* L., *Alternanthera philoxeroides* (Mart.) Griseb., *Viciahirsuta* (L.) Gray, and *Launaea aspleniifolia* (Willd.) Hook.f. The dominant weed species observed at 30 and 60 days after sowing (DAS) were *A. arvensis*, *C. rotundus,* and *V. hirsuta.* At the early crop growth stage (30 DAS), different nutrient management practices had no significant impact on weed density except on *C. rotundus* in both the years (Figure 1 and Table 1). It was found that the density of *C. rotundus* was drastically reduced by ~56% in the second year of the experimentation as compared to the first year. During the initial year of study, the densities of *A. arvensis* and *V. hirsuta* at 60 DAS were not influenced significantly with the supplementation organic manures, whereas, during the second year of study, concentrated organic manures, i.e., BSM and NC, efficiently reduced the density of *A. arvensis*. On the other hand, supplementation of N through organic manures had no impact on densities of *C. rotundus* and *V. hirsuta*.

Among the different weed management practices, application of atrazine as pre-emergence (PRE) significantly reduced the density of *A. arvensis* and *C. rotundus* at 30 DAS (Figure 1 and Table 1). The seed germination of *A. arvensis* was inhibited greatly with the application of atrazine at PRE in year 1 and complete inhibition was noticed in year 2. On the other hand, the application of atrazine at PRE had no significant effect on the density of *V. hirsuta* at 30 DAS in both years. Whereas, the integrated approach, i.e., PRE herbicide followed by mechanical weeding at 30 DAS, performed significantly better in reducing the density of *C. rotundus* and *V. hirsuta* at 60 DAS compared to the sole chemical approach. The interaction effect of different nutrient sources and weed management practices was significant on the density of *A. arvensis* at both stages (Table 1). Under the un-weeded situation, the density of *A. arvensis* was diminished by ~80% in year 2 as compared to year 1. During the investigation, the density of *A. arvensis* at both the crop growth stages was lower when concentrated organic manures, i.e., BSM and NC, were applied as N supplementation. But, such a definite trend was not observed with the density of *C. rotundus* and *V. hirsuta.*

### 2.2. Weed Dry Biomass

Like weed density, weed dry biomass at 30 DAS was also influenced slightly by the supplementation of N through different organic manures in maize (Figure 2 and Table 2). The effect of different nutrient management practices on dry biomass of various weed species at the early crop growth stage was non-significant. Whereas, during the second year of study, at the later crop growth stage, the effect was significant on biomass accumulation of *A. arvensis* and *V. hirsuta* (Table 2).

The biomass accumulation pattern by different weed flora in both years of experimentations was varied. Biomass accumulation by the dominant weeds, *A. arvensis* and *C. rotundus,* was reduced immensely in year 2 as compared to year 1, whereas an opposite trend was observed in case of *V. hirsuta*. The supplementation of N through BSM and NC significantly reduced the dry biomass of *A. arvensis* at 60 DAS in year 2 as compared to sole chemical fertilizer. Like density, also, biomass accumulation by *C. rotundus* at 60 DAS was not influenced by different nutrient management practices.

Among various weed management practices, application of atrazine at PRE significantly reduced the biomass accumulation of *A. arvensis* at 30 DAS. The sole PRE herbicide was effective in reducing the biomass of *C. rotundus* in the first year but not in the second year. The application of atrazine at PRE was not of any utility in reducing biomass accumulation by *V. hirsuta* at 30 DAS. The mechanical weeding at 30 DAS effectively lowered the biomass accumulation of *C. rotundus* for the later crop growth stage, which was not controlled through PRE herbicide alone. The lowest biomass accumulation by *V. hirsuta* was recorded with integrated weed management practice (PRE herbicide followed by mechanical weeding) in both years.

At the later growth stage of maize plant, in comparison to bulky organic manures as well as sole chemical fertilizer, supplementation of concentrated organic manures, i.e., BSM and NC, significantly reduced the dry matter accumulation of *A. arvensis* under the weedy check situation in year 2 (Table 2). In case of biomass accumulation by *V. hirsuta*, it was found that during the initial year of experimentation, application of NC was not performed well. Whereas, during the second year of experimentation, it significantly diminished the growth of *V. hirsuta* compared to the other nutrient management practices.

### 2.3. Nutrient Uptake by Weeds

Nutrients’, i.e., nitrogen (N), phosphorus (P), and potassium (K), uptake by predominated weeds of maize crop at its 60 DAS stage was assessed and the data are depicted in Table 3 and Table 4. Different nutrient sources played a significant role in nutrients’ uptake by *A. arvensis* in the second year of experimentation only, whereas the nutrients’ uptake by *C. rotundus* and *V. hirsute* at this crop growth stage was not varied statistically throughout the experimentation. As compared to sole chemical fertilizer, N supplementation through BSM and NC effectively reduced the nutrients’ uptake by *A. arvensis* in year 2. The different weed management practices had a significant role in preventing weed growth and these had a simultaneous significant impact on N, P, and K uptake by different weeds. As compared to the un-weeded situation, the application of atrazine at PRE significantly reduced the nutrients’ uptake by *A. arvensis* and *V. hirsuta*. In both years of research, sole application of atrazine at PRE (chemical approach) had no significant effect in reducing nutrients’ uptake by *C. rotundus*. On the other hand, integration of mechanical weeding with PRE herbicide effectively lowered the nutrient removal by *C. rotundus* throughout the experimentation. Irrespective of year, the interaction effect of nutrient and weed management practices had a significant impact on nutrients’ uptake by *A. arvensis* (Table 4), whereas the nutrients’ uptake by *C. rotundus* and *V. hirsute* did not differ statistically. Under the un-weeded situation, as compared to sole chemical fertilizer, application of organic manures reduced nutrient uptake by *A. arvensis* in year 1. But, during the second year of experimentation, only concentrated organic manures performed well in reducing nutrient uptake by *A. arvensis*.

### 2.4. Crop Growth

Other than weeds, supplementation of organic manures also influenced the plant height of maize during the second year of experimentation (Figure 3 and Table 5). Among the various nutrient management practices, the maximum plant height was recorded with the application of BSM for N supplementation. The crop growth rate (CGR) of maize plant was highest with sole chemical fertilization in the first year, whereas, in the second year, N supplementation through FYM produced the maximum CGR of maize. In respect of weed management practices, as compared to weedy check, both the weed management practices produced significantly taller maize plants, having higher LAI and CGR in both years. Integration of mechanical weeding with PRE herbicide produced taller maize plants with higher LAI and CGR compared to the sole herbicide application. The interaction effect of different nutrient and weed management practices exhibited significant variation in crop growth parameters during both years of experimentation. Under the weedy check situation, the higher maize plant was recorded with sole chemical fertilizer in the first year, whereas, for the second year, it was observed with supplementation of N through bulky organic manures. On the other hand, addition to organic manures had no significant impact on soil properties after one year of experimentation (Appendix A).

## 3. Discussion

The predominant weed flora in the experimental field during the maize growing season were *A. arvensis* and *C. rotundus*. At the early growth stage, the relative density of these weeds under the un-weeded situation in maize was 53% and 37%, respectively. The application of organic manures (bulky and concentrated) for N supplementation had no significant impact on weed density and biomass accumulation. But, during the second year of study at the later crop growth stage, application of concentrated organic manures, BSM and NC, were effective in reducing the density of the most dominating weed, *A. arvensis,* by 53% and 62% (weed density in RDF, RDF + BSM, and RDF + NC was 209, 98, and 79 m^−2^) respectively, whereas, the dry biomass accumulation (g m^−2^) by *A. arvensis* was curtailed at 33% with the application of BSM and NC (Figure 1 and Table 1). It might be due to the allelopathic activity of BSM and NC on weed seed germination and seedling growth. Although, no allelopathic effect was observed on seed germination and growth of maize plant. Hoagland et al. [21] reported that application of BSM, a by-product of biodiesel production, releases allelopathic phytochemicals, which played a significant role in reducing weed emergence and increasing weed seedling mortality. Similarly, the weed-suppressing ability of neem seed powder was also documented by Marley et al. 24], and they observed that the application of neem seed powder significantly reduced the emergence of *Striga*. In another lab study, Abdulla and Kumar [22] found that, the presence of glucosinolate (allelochemicals), an aqueous extract of the mustard cake, had an allelopathic impact on germination and seedling growth of greengram (*Vignaradiata*) and summer weeds. But, the allelopathic effect of concentrated organic manures was meagre in inhibiting the germination and growth of bold seeded weeds having hard seed coat or weeds propagating through tubers which are robust and hardy in nature. As a result, the application of organic manures for N supplementation had no impact on density and dry biomass of *V. hirsuta* and *C. rotundus*. But, the application of NC reduced the biomass accumulation by *V. hirsuta* at 60 DAS in the second year of study, which might be due to the repeated addition of neem cake in the same plot, which increased the concentration of allelochemicals, and such allelopathic effect was more pronounced in the second year as compared to the first year.

Application of atrazine at PRE inhibited the germination of *A. arvensis* but was unable to control emergence of weeds like *C. rotundus*, *A. philoxeroides,* and *V. hirsuta*, which might have been caused by the hardy nature of tuber, rhizome, and seed coating of these weeds. Integration of mechanical weeding with PRE herbicide effectively manages these weeds, except for *A. philoxeroides* (data not presented). The biomass accumulation (g m^−2^) by *C. rotundus* at 60 DAS was curtailed by 80% and 74% (weed biomass in chemical and integrated was 85.9 and 17.2 in year 1 and 22.4 and 5.8 in year 2, respectively) with the integration of mechanical weeding with PRE herbicide as compared to sole atrazine application in year 1 and year 2, respectively. This integration also trimmed down the biomass accumulation of *V. hirsuta* by 66% and 89% (weed biomass in chemical and integrated was 2.28 and 0.26 in year 1 and 9.04 and 3.05 in year 2, respectively) as compared to sole atrazine application in the first and second year of study, respectively (Figure 2 and Table 2). It might be due to the effect of mechanical weeding in controlling those weeds which were not usually controlled with the application of atrazine at PRE. The result is in close proximity with the findings of Rao et al. [25], who also found a lower weed dry biomass with the application of atrazine followed by hand weeding at 30 DAS in the winter maize crop at the Regional Agricultural Research Station, Lam, Guntur, Andhra Pradesh. Throughout the crop growth period, it was observed that the population and dry biomass accumulation of most dominating weeds under the un-weeded situation was reduced by 54% and 57% respectively, in the second year of experimentation as compared to the first year. This reduction was more prominent in the case of *C. rotundus*, which might have been caused by puddling performed during the previous season of rice transplanting. Practically, during puddling, the nuts of *C. rotundus* come out from the depth of the soil to the surface and usually float in the ponded water above the puddled soil, these propagating materials cannot anchor to the soil and are being exposed to scorching sunlight. As a result, subsequently, they lose their viability due to such unfavorable conditions. Thus, it has been proven to be an effective system approach for controlling noxious weeds like *C. rotundus*.

The nutrient removal is dependent on dry matter accumulation of weeds, and the weed management methods reflecting low weed dry weight had the minimum nutrient (N, P, and K) depletion. Application of BSM for N supplementation effectively reduced the growth of predominant weeds like *A. arvensis* and *C. rotundus* at 60 DAS (Figure 3). As a result of lower weed dry weight obtained in case of BSM, the N, P, and K removal was also lower than those obtained in case of other nutrient management practices. Both the weed management practices, i.e., chemical and integrated ones, trimmed down the nutrient depletion by 98% and 99% respectively, in *A. arvensis* at 60 DAS. As compared to the sole application of atrazine, the integration of mechanical weeding with PRE herbicide restricted the N, P, and K depletion by *C. rotundus* by 77%. Sunitha et al. [26] also observed that PRE application of atrazine followed by hand weeding at 30 DAS lessened the weed N, P, and K uptake by 42.3%, 54.0%, and 46.2% respectively, over the weedy check.

Initially, application of N through a sole inorganic source increased the plant growth of maize, whereas during the second year, better growth and yield were recorded from the plots that received concentrated organic manures (BSM and NC) than that of sole inorganic fertilizer-applied plots. Inorganic fertilizers are capable of supplying nutrients more rapidly to the plants in adequate amounts, which reflected the positive impact on plant growth parameters, but the organic manure releases plant nutrients slowly and steadily to the crops, resulting in its better performance than the application of plant nutrient through inorganic fertilizers in subsequent years. The nutrients present in the manures remain in the soil as a reserve and become available to the crop bit by bit through mineralization. This result is in agreement with the statements of Xu et al. [27] and Srivastava et al. [28]. The addition of concentrated organic manures (BSM and NC) reduced the growth and nutrient (N, P, and K) depletion by weeds, and for that reason, hastened the growth of maize plant, which ultimately resulted in higher maize grain yield. Nagavani and Subbian [29] found that the application of 50% N through inorganic fertilizer +50% N through poultry manure effectively reduced the weed density and growth as well as significantly increased the grain and stover yield of maize. The integration of mechanical weeding with PRE-herbicide was more effective in reducing weed growth as well as nutrient depletion by weeds and eventually accelerated the growth of the maize plant. Sunitha et al. [26] and Sanodiya et al. [30] also found similar results (e.g., yield enhancement) with integrated weed management in maize.

## 4. Materials and Methods

### 4.1. Experimental Location

The field experiment was conducted in the winter seasons of 2014–2015 (Year 1) and 2015–2016 (Year 2) in a farmer’s field situated at Uttar Chandamari village, Muratipur, Nadia, West Bengal, India (88°27 N latitude and 22°59 E longitude, with the elevation of 7.9 m above the mean sea level). The agro-meteorological parameters were recorded at the Meteorological Observatory (approximately 1.0 km away from the field site) under the All India Coordinated Research Project on Agrometeorology, BCKV, Kalyani, West Bengal, India. The climate of the experimental site is of sub-tropical humid type with an average annual rainfall of 1400 mm, mostly precipitated from June to September, and the mean temperature ranges from 6.9 to 40.4 °C. During the experimental season, the maximum and minimum temperature fluctuated between 9.6 and 37.3 °C in 2014−2015, and 35.1 and 9.3 °C in 2015−2016. Generally, the temperature falls during the month of November and remains low until February, favoring the growth and development of maize. The maximum and minimum relative humidity prevailed between 89% and 34% in 2014–2015, and 95% and 42% in 2015−2016. The rainfall during the experimental period (November to March) was 24.2 and 73.9 mm in 2014−2015 and 2015−2016, respectively. The soil of the experimental field was typically of Gangetic alluvium (*Entisol*) type, clay loam in texture, with a pH of 6.27, electrical conductivity (dSm^−1^) of 0.19, and medium in organic carbon (0.52%), low in available N (215 kg N ha^−1^), high in available P (36.3 kg ha^−1^), and medium in available K (173 kg ha^−1^).

### 4.2. Experimental Setup and Crop Management

The experiment was laid out in a replicated factorial randomized block design having two factors: nutrient management and weed management. The nutrient management factor having five levels (sole chemical fertilizer, integration of chemical fertilizer with bulky organic manures, i.e., farmyard manure (FYM) or vermicompost (VC), and integration of chemical fertilizer with concentrated organic manures, i.e., *Brassicaceous* seed meal (BSM) or neem cake (NC), for 25% of recommended N in maize), and the weed regimes factor with three levels, i.e., weedy check, sole chemical approach (atrazine 1000 g ha^−1^ at 2 days after sowing (DAS)), and integrated weed management (atrazine 1000 g ha^−1^ at 2 DAS followed by mechanical weeding at 30 DAS). The recommended dose of fertilizer (RDF) for hybrid maize was 200-60-60 kg N-P_2_O_5_-K_2_O ha^−1^ [14]. The nutrients were applied through urea (46% N), single super phosphate (16% P_2_O_5_), and muriate of potash (60% K_2_O). Total P and K fertilizers were applied to the soil prior to sowing of maize crop. The N fertilizer was applied in three splits, i.e., ½ N before sowing, ¼ N at the knee-height stage, and ¼ N at the pre-tasselling stage. All the organic manures for 25% recommended N were applied before sowing of the crop (two days before final land preparation). The chemical composition of organic fertilizers is presented in Appendix A. By adding the organic manures, approximately 10–21 kg P_2_O_5_ ha^−1^ and 12–36 kg K_2_O ha^−1^ were also added. The herbicides were applied with a knapsack sprayer (Model: AGM/001, ASPEE Sprayers and Farm Mechanized Equipment, Mumbai, India) of 16 liters capacity with flat fan nozzles, and the spray volume was maintained at 500 L ha^−1^. The mechanical weeding was performed with a wheel hoe at 30 DAS. In the weedy control, no weeding was done.

The hybrid maize cultivar (cv. P-3396 of Pioneer Seed P Ltd., India) was sown on the tenth and seventh day of November 2014 and 2015 respectively, in rows 60 cm apart at a depth of 3–5 cm, and the plant-to-plant spacing was 30 cm. Individual plots were 7.2 m wide and 3 m long and were separated from adjacent plots by a 1.0 m wide plant-free border. All general agronomic practices like irrigation, intercultural operations, harvesting, and necessary plant protection measures for the region, were followed as suggested by Sharma and Das [15] and Ray et al. [31]. After sowing, irrigation was applied for uniform germination, and the field was visited regularly to check the irrigation demand of the crop as per the conditions of soil and crop. During the first year of experimentation, five irrigations were given during seeding, knee-height, tasseling, silking, and grain-filling stages, whereas in the second year, four irrigations were provided during seeding, knee-height, tasseling, and grain-filling stages. In each plot, the second rows on either side were marked for destructive plant sampling and for the recording of other biometric observations. The middle eight rows were marked for the determination of yield. The maize crop was harvested manually with a sickle at a height of 10–15 cm from ground level on 22 and 20 of March 2015 and 2016, respectively.

### 4.3. Biometric Measurements

For weed density and weed dry biomass accumulation at 30 and 60 DAS, four permanent quadrats (1 × 1 m) were earmarked in each plot after maize sowing. Weed density was measured as the number of weeds per unit area from two permanent quadrats at 30 and 60 DAS. During each observation period, weeds from two quadrats were cut at ground level for measuring weed dry weight. For taking dry weight, the destructed weed samples were first washed in clean tap water, then sun-dried and hot-air oven-dried at 70 °C for 48 h and weighed [10,19].

Ten maize plants were selected randomly from each plot of net plot area and heights were taken from the ground level to the tip of the plant, and for plant dry matter, accumulation of five plants were cut to ground level from the area specified for destructive plant sampling. The plant samples were dried in an electrical oven at 70 °C temperature for 48 h. From these observations, dry matter weight was worked out in g/plant. Crop growth rate, defined as the increase in dry weight of plant material per unit area of land per unit change of time, was calculated with the following formula [20]:(1)$$\mathrm{CGR}=\frac{W_{2}-W_{1}}{t_{2}-t_{1}}\;{\mathrm{gm}}^{-2}{\mathrm{day}}^{-1}$$
where, W_1_ = initial dry weight per unit area at t_1_ time (initial sampling date), and W_2_ = final dry weight per unit area at t_2_ time (final date of sampling).

For leaf area index (LAI) calculation, leaf laminae were separated from sampled plants for recording area and dry weight. For determining the LAI, leaf punch or leaf cut was taken. For maize plant, leaf punches were taken with the help of a punching core of the known area. Then, the dry weight of those samples was recorded, and the area-weight relationship was calculated. By using this relationship, the leaf area of the sampled leaves was worked out. The LAI was calculated with the following formula given by Watson [32]:(2)$$\mathrm{LAI}\;=\frac{\mathrm{Area}\;\mathrm{of}\;\mathrm{total}\;\mathrm{number}\;\mathrm{of}\;\mathrm{leaves}\;\left({\mathrm{cm}}^{2}\right)}{\mathrm{The}\;\mathrm{ground}\;\mathrm{area}\;\mathrm{from}\;\mathrm{where}\;\mathrm{leaf}\;\mathrm{samples}\;\mathrm{were}\;\mathrm{collected}\;\left({\mathrm{cm}}^{2}\right)}$$

Grain yield of maize was estimated from the demarked net plot area and after harvesting, all the cobs were removed from them. Grains were deshelled from the cob (de-husked) and then dried properly to reduce the moisture content to 14.0%. Weight of grains was recorded as kg plot^–1^ and then converted to t ha^–1^ [2].

### 4.4. Analytical Procedures

The total nitrogen, available phosphorus, and potassium content of organic manures, i.e., Farm Yard Manure (FYM), vermicompost (VC), and *Brassicaceous* seed meal (BSM), were determined in each year before applying to the soil following standard analytical methodologies, e.g., Subbiah and Asija [33] for N, Olsen et al. [34] for P, and Brown and Warncke [35] for K. Crop and individual weed species samples from each treatment were collected, oven-dried, and ground for analyzing total recoveries of N, P, and K. Nitrogen was estimated by the micro-Kjeldahl method. For determination of P and K content, plant material was digested in tri-acid (HNO_3_:H2SO_4_:HClO_4_ = 10:1:4) [36] and estimated by spectrophotometer and flame photometer, respectively.

### 4.5. Experimental Design and Statistical Analysis

The experiment was conducted in a replicated factorial randomized block design with three replications. The statistical analysis of the experimental data was done year-wise using SAS Windows Version 9.3 (SAS Institute Inc., Cary, NC, USA, www.sas.com). The Excel software (version 2016, Microsoft Inc., WA, USA, www.microsoft.com) was used to draw graphs and figures.

## 5. Conclusions

From the study, it was concluded that the integration of chemical fertilizer with concentrated organic manures helps to manage weeds by restricting growth and nutrient removal by major dominating weeds. The addition of organic manures also enhanced the growth and yield of maize crop. The concentrated organic manures as a source of nutrients along with herbicide and mechanical weeding augmented the growth and yield of maize crop by suppressing weeds. Thus, integrated weed and nutrient management strategies might be a potential tool for sustainable production of hybrid maize for smallholder farmers of eastern Indo-Gangetic plains.

## Figures and Tables

**Figure 1 molecules-25-04924-f001:**
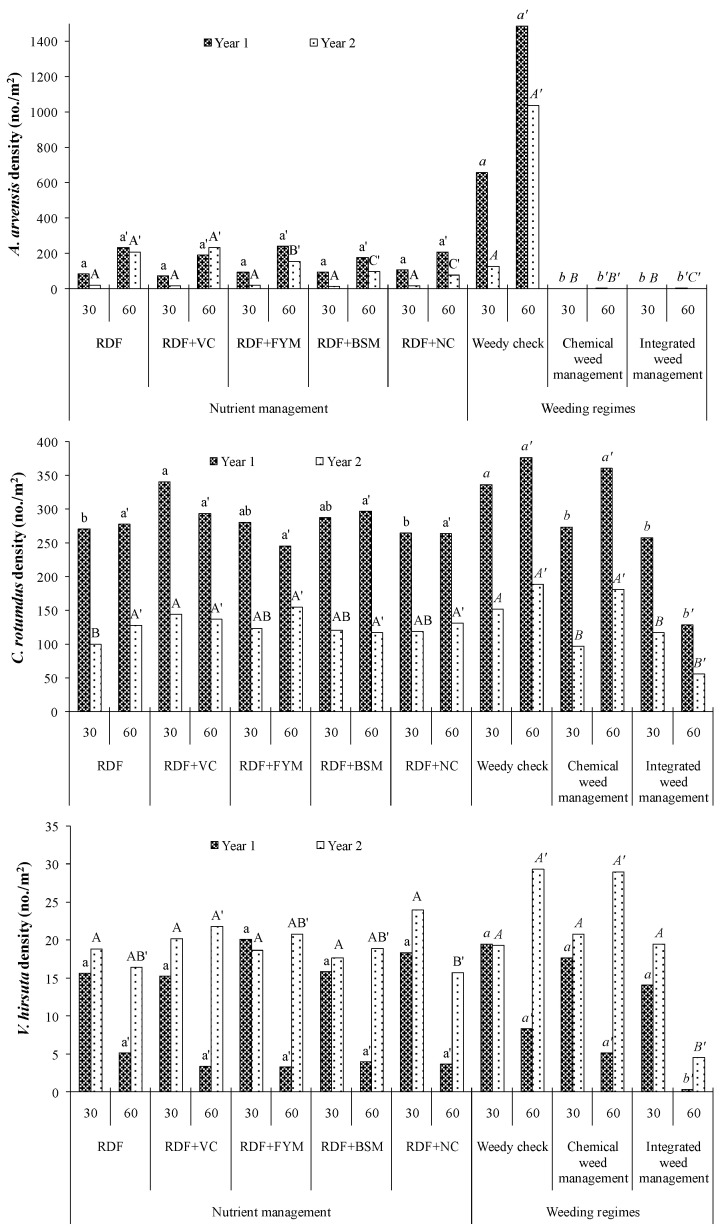
Effect of different nutrient managements and weeding regimes on weed density at 30 and 60 days after seeding (DAS) in hybrid maize. RDF (recommended dose of fertilizer): 100% recommended dose (RD) of NPK through fertilizer, RDF + VC: 25% RDN through VC, RDF + FYM: 25% RDN through FYM, RDF + BSM: 25% RDN through BSM, RDF + NC: 25% RDN through NC, Weedy check: Un-weeded control, Chemical weed management: Atrazine 1000 g/ha at 2 DAS, Integrated weed management: Atrazine 1000 g ha^−1^ at 2 DAS followed by mechanical weeding at 30 DAS. Effects of nutrient source recorded at 30 DAS in year 1 indicated with the lower case of letter (a, b or ab) and indicated with upper case of letter (A, B or AB) in year 2; however data recorded at 60 DAS indicate as lower case with (′) (a’ or b’) in year 1 and upper case with (′) (A’, B’ or AB’) in year 2; while effect of weed management in year 1 as italic lower case (a, b or ab), in year 2, italic upper case (A, B or AB) at 30 DAS, as italic lower case with (′) (a’ or b’) in year 1 and italic upper case with (′) (A’, B’ or AB’) in year 2 at 60 DAS.

**Figure 2 molecules-25-04924-f002:**
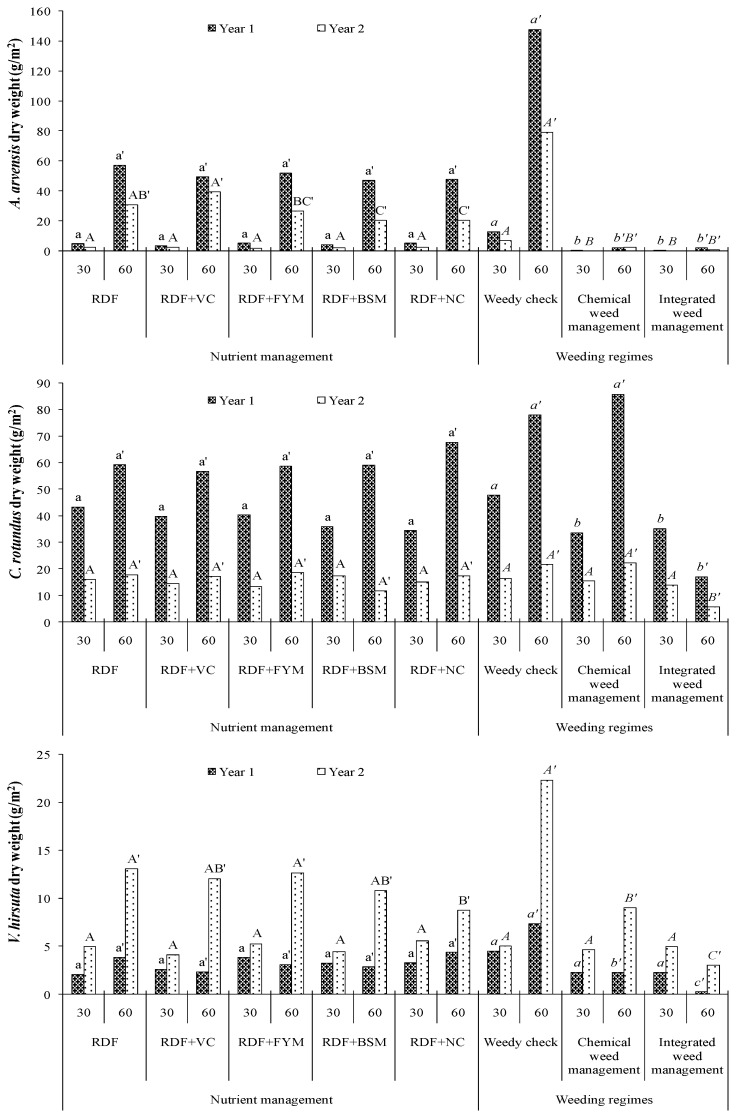
Effect of different nutrient managements and weeding regimes on weed dry weight (g/m^2^) at both 30 and 60 DAS in hybrid maize. RDF: 100% recommended dose (RD) of NPK through fertilizer, RDF + VC: 25% RDN through VC, RDF + FYM: 25% RDN through FYM, RDF + BSM: 25% RDN through BSM, RDF + NC: 25% RDN through NC, Weedy check: Un-weeded control, Chemical weed management: Atrazine 1000 g ha^−1^ at 2 DAS, Integrated weed management: Atrazine 1000 g/ha at 2 DAS followed by mechanical weeding at 30 DAS. Effects of nutrient source recorded at 30 DAS in year 1 indicated with the lower case of letter (a or b) and indicated with upper case of letter (A or B) in year 2; however data recorded at 60 DAS indicate as lower case with (′) (a’ or b’) in year 1 and upper case with (′) (A’, B’ or AB’) in year 2; while effect of weed management in year 1 as italic lower case, in year 2, italic upper case at 30 DAS, as italic lower case with (′) (a’ or b’) in year 1 and italic upper case with (′) (A’, B’ or AB’) in year 2 at 60 DAS.

**Figure 3 molecules-25-04924-f003:**
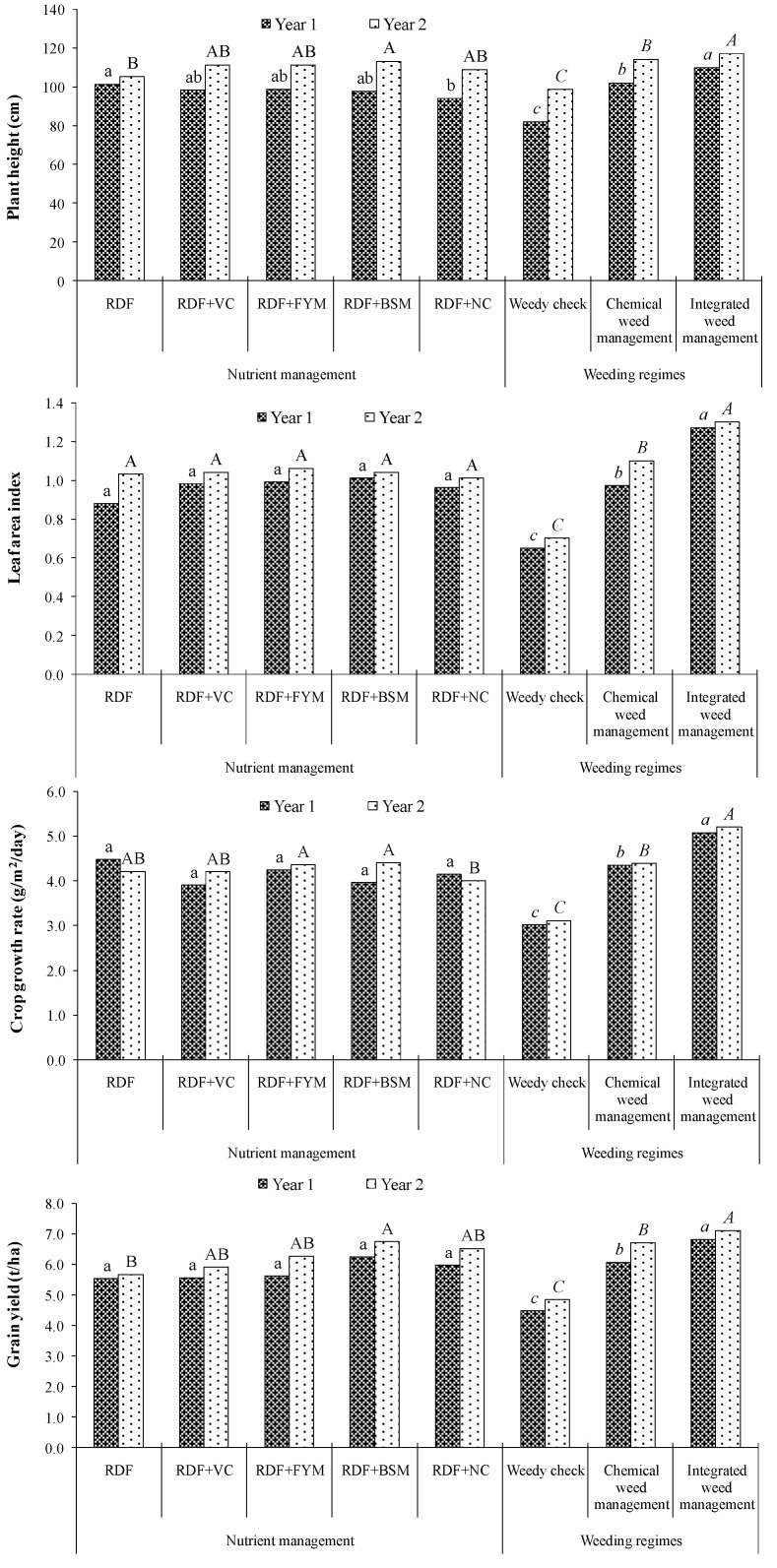
Effect of different nutrient managements and weeding regimes on plant height, leaf area index (LAI) recorded at 60 DAS, crop growth rate (CGR) recorded at 60–90 DAS, and grain yield of hybrid maize. RDF: 100% recommended dose (RD) of NPK through fertilizer, RDF + VC: 25% RDN through VC, RDF + FYM: 25% RDN through FYM, RDF + BSM: 25% RDN through BSM, RDF + NC: 25% RDN through NC, Weedy check: Un-weeded control, Chemical weed management: Atrazine 1000 g/ha at 2 DAS, Integrated weed management: Atrazine 1000 g ha^−1^ at 2 DAS followed by mechanical weeding at 30 DAS. Effects of nutrient source in year 1 as lower case (a, b or ab), in year 2, upper case (A, B or AB), whereas effect of weed management in year 1 as italic lower case (*a, b*
*or c*), in year 2 (*A, B*
*or C*), italic upper case.

**Table 1 molecules-25-04924-t001:** Interaction effect of different nutrient sources and weeding regimes on weed density (number m^−2^) in hybrid maize.

Treatments Combination		Weed Density (number m^−2^)									
		*A. arvensis*				*C. rotundus*				*V. hirsuta*	
		30 DAS		60 DAS		30 DAS		60 DAS		60 DAS	
		Year 1	Year 2	Year 1	Year 2	Year 1	Year 2	Year 1	Year 2	Year 1	Year 2
Weedy Check	RDF	24.0 (574)	12.6 (159)	41.2 (1699)	38.3 (1469)	18.3 (333)	10.7 (114)	21.2 (448)	13.5 (182)	3.09 (9.06)	5.48 (29.53)
	RDF + VC	21.4 (459)	11.2 (124)	36.9 (1363)	36.7 (1346)	19.5 (379)	14.4 (206)	19.9 (395)	14.1 (199)	3.05 (8.83)	5.64 (31.30)
	RDF + FYM	26.7 (714)	12.2 (148)	40.7 (1658)	32.4 (1049)	19.5 (381)	12.3 (151)	20.5 (421)	14.2 (201)	2.73 (6.95)	5.71 (32.05)
	RDF + BSM	26.9 (721)	9.8 (95)	34.7 (1203)	28.3 (802)	17.1 (293)	13.5 (180)	17.5 (304)	13.4 (179)	2.41 (5.33)	5.39 (28.60)
	RDF + NC	29.4 (861)	11.1 (123)	39.2 (1537)	25.4 (643)	17.4 (302)	10.8 (116)	18.1 (328)	13.6 (184)	3.62 (12.57)	5.09 (25.45)
Chemical Weed Management	RDF	2.05 (3.69)	0.71 (0)	2.6 (6.5)	4.3 (18.4)	16.6 (274)	8.2 (66)	18.2 (332)	12.6 (158)	2.98 (8.35)	5.21 (26.66)
	RDF + VC	2.19 (4.30)	0.71 (0)	2.7 (7.0)	5.6 (31.0)	19.4 (375)	9.9 (97)	19.8 (391)	13.7 (187)	2.19 (4.30)	5.31 (27.65)
	RDF + FYM	1.46 (1.64)	0.71 (0)	2.6 (6.3)	3.2 (9.5)	15.2 (229)	7.9 (62)	17.2 (294)	15.1 (229)	2.41 (5.33)	5.73 (32.31)
	RDF + BSM	1.07 (0.66)	0.71 (0)	3.0 (8.4)	0.7 (0.0)	16.6 (275)	12.4 (153)	21.5 (462)	12.4 (152)	2.51 (5.79)	5.20 (26.54)
	RDF + NC	1.07 (0.66)	0.71 (0)	1.3 (1.2)	0.7 (0.0)	15.2 (230)	11.1 (123)	18.4 (339)	13.5 (182)	1.83 (2.85)	5.71 (32.15)
Integrated Weed Management	RDF	1.66 (2.25)	0.71 (0)	2.0 (3.3)	0.7 (0.0)	14.6 (211)	11.3 (126)	10.6 (113)	8.0 (63)	1.07 (0.66)	1.66 (2.25)
	RDF + VC	2.05 (3.69)	0.71 (0)	1.8 (2.8)	3.5 (12.1)	16.6 (275)	11.8 (138)	11.8 (138)	7.4 (55)	0.71 (0.00)	3.22 (9.87)
	RDF + FYM	1.07 (0.66)	0.71 (0)	3.2 (9.8)	1.9 (3.0)	15.6 (244)	13.0 (169)	9.4 (88)	8.1 (65)	0.71 (0.00)	2.41 (5.33)
	RDF + BSM	1.46 (1.64)	0.71 (0)	2.2 (4.4)	0.7 (0.0)	17.3 (298)	7.2 (52)	12.8 (162)	6.7 (44)	1.44 (1.58)	2.63 (6.42)
	RDF + NC	0.71 (0.00)	0.71 (0)	2.8 (7.4)	0.7 (0.0)	16.3 (265)	10.8 (117)	12.3 (150)	7.3 (53)	0.71 (0.00)	1.29 (1.17)
SEm ±		2.84	0.68	2.09	1.06	1.10	1.16	1.79	1.30	0.53	0.37
CD (*p* ≤ 0.05)		8.22	1.98	6.06	3.06	NS	3.36	5.17	3.77	1.55	1.07

RDF: 100% recommended dose (RD) of NPK through fertilizer, RDF + VC: 25% RDN through VC, RDF + FYM: 25% RDN through FYM, RDF + BSM: 25% RDN through BSM, RDF + NC: 25% RDN through NC, Weedy check: Un-weeded control, Chemical weed management: Atrazine 1000 g/ha at 2 DAS, Integrated weed management: Atrazine 1000 g ha^−1^ at 2 DAS followed by mechanical weeding at 30 DAS. SEm, Standard error of mean; CD, Critical difference; NS, Non-significant. Data given in the parenthesis were original and subjected to square root transformation ($\sqrt{x+0.5}$).

**Table 2 molecules-25-04924-t002:** Interaction effect of different nutrient managements and weeding regimes on weed dry weight (g m^−2^) in hybrid maize.

Treatments Combination		Weed dry Weight (g m^−2^)							
		*A. arvensis*				*C. rotundus*		*V. hirsuta*	
		30 DAS		60 DAS		30 DAS		60 DAS	
		Year 1	Year 2	Year 1	Year 2	Year 1	Year 2	Year 1	Year 2
Weedy check	RDF	13.23	8.09	168.4	88.7	47.4	17.0	7.96	21.85
	RDF + VC	8.89	7.69	145.8	108.8	49.2	14.9	5.13	25.33
	RDF + FYM	14.11	5.78	150.4	76.3	62.7	15.5	7.69	21.58
	RDF + BSM	12.44	6.88	135.5	61.4	37.2	19.6	5.63	23.22
	RDF + NC	15.43	7.10	139.1	61.4	42.6	15.8	10.56	19.81
Chemical weed management	RDF	0.64	0.00	2.3	3.7	35.6	13.4	3.29	15.33
	RDF + VC	0.94	0.00	1.5	5.8	38.2	13.2	1.94	6.86
	RDF + FYM	0.58	0.00	2.4	2.5	28.2	13.4	1.65	10.91
	RDF + BSM	0.64	0.00	3.7	0.0	36.6	21.7	1.97	6.25
	RDF + NC	0.24	0.00	1.4	0.0	29.8	15.9	2.56	5.82
Integrated weed management	RDF	0.71	0.00	1.1	0.0	47.5	17.9	0.32	2.07
	RDF + VC	0.91	0.00	1.3	3.5	32.4	15.7	0.00	4.05
	RDF + FYM	1.11	0.00	3.5	0.9	30.2	11.7	0.00	5.53
	RDF + BSM	0.05	0.00	2.4	0.0	34.6	11.5	0.99	2.96
	RDF + NC	0.00	0.00	2.4	0.0	31.5	13.9	0.00	0.67
SEm ±		2.41	0.92	16.7	5.5	5.7	3.3	1.65	2.30
CD (*p* ≤ 0.05)		6.99	2.67	48.4	15.9	16.4	NS	4.78	6.65

RDF: 100% recommended dose (RD) of NPK through fertilizer, RDF + VC: 25% RDN through VC, RDF + FYM: 25% RDN through FYM, RDF + BSM: 25% RDN through BSM, RDF + NC: 25% RDN through NC, Weedy check: Un-weeded control, Chemical weed management: Atrazine 1000 g/ha at 2 DAS, Integrated weed management: Atrazine 1000 g/ha at 2 DAS followed by mechanical weeding at 30 DAS. SEm, Standard error of mean; CD, Critical difference; NS, Non-significant.

**Table 3 molecules-25-04924-t003:** Effect of different nutrient managements and weeding regimes on nutrient uptake by major weeds (kg ha^−1^) at 60 DAS in hybrid maize.

Treatments	Nutrient Uptake by Major Weeds (kg ha^−1^) at 60 DAS																	
	*A. arvensis*						*C. rotundus*						*V. hirsuta*					
	Nitrogen		Phosphorus		Potassium		Nitrogen		Phosphorus		Potassium		Nitrogen		Phosphorus		Potassium	
	Year 1	Year 2	Year 1	Year 2	Year 1	Year 2	Year 1	Year 2	Year 1	Year 2	Year 1	Year 2	Year 1	Year 2	Year 1	Year 2	Year 1	Year 2
**Nutrient Management**																		
RDF	17.36	9.34	2.31	1.24	1.44	0.78	16.04	4.86	1.99	0.60	1.48	0.45	0.77	2.62	0.15	0.50	0.07	0.25
RDF + FYM	15.02	11.94	2.00	1.59	1.25	0.99	15.34	4.66	1.91	0.58	1.41	0.43	0.47	2.42	0.09	0.46	0.05	0.23
RDF + VC	15.79	8.05	2.10	1.07	1.31	0.67	15.88	5.04	1.97	0.63	1.46	0.46	0.62	2.54	0.12	0.48	0.06	0.24
RDF + BSM	14.31	6.21	1.90	0.82	1.19	0.52	16.00	3.18	1.99	0.40	1.47	0.29	0.57	2.16	0.11	0.41	0.06	0.21
RDF + NC	14.45	6.20	1.92	0.82	1.20	0.52	18.32	4.74	2.28	0.59	1.69	0.44	0.88	1.75	0.17	0.33	0.08	0.17
SEm ±	2.93	0.96	0.39	0.13	0.24	0.08	3.07	0.81	0.38	0.10	0.28	0.08	0.19	0.27	0.04	0.05	0.02	0.03
CD (*p* ≤ 0.05)	8.48	2.79	1.13	0.37	0.70	0.23	NS	NS	NS	NS	NS	NS	NS	NS	NS	NS	NS	NS
**Weeding Regimes**																		
Weedy check	44.83	24.05	5.96	3.20	3.73	2.00	21.08	5.89	2.62	0.73	1.94	0.54	1.48	4.48	0.28	0.85	0.14	0.43
Chemical weed management	0.69	0.72	0.09	0.10	0.06	0.06	23.20	6.04	2.89	0.75	2.14	0.56	0.46	1.81	0.09	0.35	0.04	0.17
Integrated weed management	0.65	0.26	0.09	0.04	0.05	0.02	4.66	1.56	0.58	0.19	0.43	0.14	0.05	0.61	0.01	0.12	0.01	0.06
SEm ±	2.27	0.75	0.30	0.10	0.19	0.06	2.38	0.63	0.30	0.08	0.22	0.06	0.15	0.21	0.03	0.04	0.01	0.02
CD (*p* ≤ 0.05)	6.57	2.16	0.87	0.29	0.55	0.18	6.90	1.83	0.86	0.23	0.64	0.17	0.43	0.60	0.08	0.11	0.04	0.06

RDF: 100% recommended dose (RD) of NPK through fertilizer, RDF + VC: 25% RDN through VC, RDF + FYM: 25% RDN through FYM, RDF + BSM: 25% RDN through BSM, RDF + NC: 25% RDN through NC, Weedy check: Un-weeded control, Chemical weed management: Atrazine 1000 g/ha at 2 DAS, Integrated weed management: Atrazine 1000 g ha^−1^ at 2 DAS followed by mechanical weeding at 30 DAS. SEm, Standard error of mean; CD, Critical difference; NS, Non-significant.

**Table 4 molecules-25-04924-t004:** Interaction effect of different nutrient managements and weeding regimes on nutrient uptake (kg ha^−1^) by Major Weeds at 60 DAS in Hybrid Maize.

Treatments Combination		Nutrient Uptake by Major Weeds (kg ha^−1^) at 60 DAS																	
		*A. arvensis*						*C. rotundus*						*V. hirsuta*					
		Nitrogen		Phosphorus		Potassium		Nitrogen		Phosphorus		Potassium		Nitrogen		Phosphorus		Potassium	
		Year 1	Year 2	Year 1	Year 2	Year 1	Year 2	Year 1	Year 2	Year 1	Year 2	Year 1	Year 2	Year 1	Year 2	Year 1	Year 2	Year 1	Year 2
Weedy check	RDF	51.05	26.91	6.79	3.58	4.24	2.24	24.12	6.69	3.00	0.83	2.22	0.62	1.59	4.37	0.30	0.83	0.15	0.42
	RDF + VC	44.22	32.99	5.88	4.38	3.68	2.74	19.32	5.81	2.40	0.72	1.78	0.54	1.03	5.07	0.20	0.97	0.10	0.49
	RDF + FYM	45.61	23.15	6.06	3.08	3.79	1.92	20.68	6.03	2.57	0.75	1.91	0.56	1.54	4.32	0.29	0.82	0.15	0.42
	RDF + BSM	41.08	18.62	5.46	2.47	3.41	1.55	20.29	4.79	2.52	0.60	1.87	0.44	1.13	4.65	0.22	0.89	0.11	0.45
	RDF + NC	42.19	18.61	5.61	2.47	3.51	1.55	21.00	6.15	2.61	0.76	1.94	0.57	2.11	3.97	0.40	0.76	0.20	0.38
Chemical weed management	RDF	0.69	1.12	0.09	0.15	0.06	0.09	20.37	6.41	2.53	0.80	1.88	0.59	0.66	3.07	0.13	0.59	0.06	0.30
	RDF + VC	0.45	1.76	0.06	0.23	0.04	0.15	23.07	7.10	2.87	0.88	2.13	0.65	0.39	1.37	0.07	0.26	0.04	0.13
	RDF + FYM	0.73	0.74	0.10	0.10	0.06	0.06	24.95	6.89	3.10	0.86	2.30	0.63	0.33	2.18	0.06	0.42	0.03	0.21
	RDF + BSM	1.13	0.00	0.15	0.00	0.09	0.00	23.25	3.73	2.89	0.46	2.14	0.34	0.40	1.25	0.08	0.24	0.04	0.12
	RDF + NC	0.43	0.00	0.06	0.00	0.04	0.00	24.38	6.07	3.03	0.75	2.25	0.56	0.51	1.16	0.10	0.22	0.05	0.11
Integrated weed management	RDF	0.34	0.00	0.04	0.00	0.03	0.00	3.62	1.49	0.45	0.19	0.33	0.14	0.06	0.41	0.01	0.08	0.01	0.04
	RDF + VC	0.39	1.06	0.05	0.14	0.03	0.09	3.63	1.08	0.45	0.13	0.33	0.10	0.00	0.81	0.00	0.15	0.00	0.08
	RDF + FYM	1.05	0.26	0.14	0.03	0.09	0.02	2.00	2.21	0.25	0.27	0.18	0.20	0.00	1.11	0.00	0.21	0.00	0.11
	RDF + BSM	0.73	0.00	0.10	0.00	0.06	0.00	4.45	1.04	0.55	0.13	0.41	0.10	0.20	0.59	0.04	0.11	0.02	0.06
	RDF + NC	0.74	0.00	0.10	0.00	0.06	0.00	9.59	2.01	1.19	0.25	0.88	0.19	0.00	0.13	0.00	0.03	0.00	0.01
SEm ±		5.07	1.67	0.67	0.22	0.42	0.14	5.32	1.41	0.66	0.18	0.49	0.13	0.33	0.46	0.06	0.09	0.03	0.04
CD (*p* ≤ 0.05)		14.68	4.83	1.95	0.64	1.22	0.40	NS	NS	NS	NS	NS	NS	NS	NS	NS	NS	NS	NS

RDF: 100% recommended dose (RD) of NPK through fertilizer, RDF + VC: 25% RDN through VC, RDF + FYM: 25% RDN through FYM, RDF + BSM: 25% RDN through BSM, RDF + NC: 25% RDN through NC, Weedy check: Un-weeded control, Chemical weed management: Atrazine 1000 g/ha at 2 DAS, Integrated weed management: Atrazine 1000 g ha^−1^ at 2 DAS followed by mechanical weeding at 30 DAS. SEm, Standard error of mean; CD, Critical difference; NS, Non-significant.

**Table 5 molecules-25-04924-t005:** Interaction effect of different nutrient managements and weeding regimes on plant height, leaf area index (LAI) at 60 DAS, crop growth rate (CGR) at 60–90 DAS, and grain yield of hybrid maize at harvest.

Treatments Combination		Plant Height (cm)		LAI		Crop Growth Rate (gm^−2^day^−1^)		Grain Yield (t ha^−1^)	
		Year 1	Year 2	Year 1	Year 2	Year 1	Year 2	Year 1	Year 2
Weedy check	RDF	87.8	93.3	0.70	0.69	3.10	3.10	4.30	4.39
	RDF + VC	78.2	100.9	0.60	0.61	2.78	2.87	4.17	4.26
	RDF + FYM	84.8	106.7	0.67	0.76	3.38	3.46	4.36	5.37
	RDF + BSM	84.7	96.0	0.70	0.73	2.89	3.15	4.93	5.04
	RDF + NC	75.0	96.3	0.59	0.73	2.88	2.93	4.61	5.12
Chemical weed management	RDF	107.5	105.9	0.85	1.01	4.85	4.57	5.85	5.99
	RDF + VC	105.7	117.0	1.05	1.20	3.86	4.59	6.15	6.61
	RDF + FYM	101.0	113.8	0.97	1.09	4.15	4.25	5.94	6.77
	RDF + BSM	96.0	122.3	0.94	1.16	4.36	4.65	5.96	7.06
	RDF + NC	100.8	112.3	1.02	1.04	4.49	3.86	6.41	7.07
Integrated weed management	RDF	109.2	116.7	1.08	1.39	5.46	4.93	6.44	6.60
	RDF + VC	111.5	116.2	1.28	1.30	5.04	5.12	6.35	6.82
	RDF + FYM	110.5	113.3	1.34	1.32	5.20	5.33	6.55	6.65
	RDF + BSM	112.3	121.4	1.40	1.24	4.60	5.37	7.83	8.07
	RDF + NC	106.2	118.1	1.25	1.26	5.04	5.19	6.85	7.36
SEm ±		3.40	3.88	0.12	0.10	0.21	0.16	0.33	0.34
CD (*p* ≤ 0.05)		9.84	11.23	0.34	0.30	0.61	0.47	0.97	0.99

RDF: 100% recommended dose (RD) of NPK through fertilizer, RDF + VC: 25% RDN through VC, RDF + FYM: 25% RDN through FYM, RDF + BSM: 25% RDN through BSM, RDF + NC: 25% RDN through NC, Weedy check: Un-weeded control, Chemical weed management: Atrazine 1000 g/ha at 2 DAS, Integrated weed management: Atrazine 1000 g ha^−1^ at 2 DAS followed by mechanical weeding at 30 DAS. SEm, Standard error of mean; CD, Critical difference; NS, Non-significant.

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
