# Peer review of "Nutrients Supplementation through Organic Manures Influence the Growth of Weeds and Maize Productivity"

_molecules, 2020, doi:10.3390/molecules25214924_

Round 1

Reviewer 1 Report

I appreciated the more focused conclusions, that clearly resume the most significant outcomes of the work.

Reviewer 2 Report

Although the authors have indicated in their cover letter that they have modified the figures, Figures 1-3 have not been updated from the previous version.

Because the experimental design includes 5 nutrient managements and 3 weed managements (showed somehow only in tables) Authors must show the data for all 15 treatments in the Figures e.g. as follow for the X axis:

30

60

30

60

30

60

30

60

30

60

30

60

30

60

30

60

30

60

30

60

30

60

30

60

30

60

30

60

30

60

RDF

RDF+VC

RDF+FYM

RDF+BSM

RDF+NC

RDF

RDF+VC

RDF+FYM

RDF+BSM

RDF+NC

RDF

RDF+VC

RDF+FYM

RDF+BSM

RDF+NC

Weedy

Chemical

Integrated